# Strong Root System Enhances Waterlogging Resilience in Barley (*Hordeum vulgare*) at the Early Stage Stress

**DOI:** 10.3390/plants15010134

**Published:** 2026-01-02

**Authors:** Yu Tian, Li Cao, Yuening Xin, Liang Zhu, Zhenxiang Zhou, Baojian Guo, Chao Lv, Juan Zhu, Rugen Xu, Feifei Wang

**Affiliations:** 1Jiangsu Key Laboratory of Crop Genomics and Molecular Breeding, Key Laboratory of Plant Functional Genomics of the Ministry of Education, Jiangsu Key Laboratory of Crop Genetics and Physiology, Agricultural College of Yangzhou University, Yangzhou 225009, China; 15138619065@163.com (Y.T.); 17205235170@163.com (L.C.); xin_yuening@outlook.com (Y.X.); zl2269096164@163.com (L.Z.); bjguo@yzu.edu.cn (B.G.); clv@yzu.edu.cn (C.L.); 007670@yzu.edu.cn (J.Z.); 2Jiangsu Co-Innovation Center for Modern Production Technology of Grain Crops, Yangzhou University, Yangzhou 225009, China; 3East China Agri-Tech Center of Chinese Academy of Agricultural Sciences, Suzhou 215000, China; zhouzhenxiang@ecs-caas.cn

**Keywords:** waterlogging, barley, root morphology, ABA, GA, ACC, gene expression

## Abstract

Barley (*Hordeum vulgare* L.) is resistant to salt, drought and low temperature stress but sensitive to waterlogging stress. For now, little is known in waterlogging stress about the molecular mechanisms underlying the regulation of hormonal pathways in barley roots. In this study, the effects of waterlogging stress on 33 genotypes during the whole growth stages were comprehensively assessed. Then, the root morphology, hormone content and relative root development gene expressions were investigated in the most waterlogging-sensitive and -tolerant genotypes. We found waterlogging-tolerant genotypes (four representative genotypes) increased root length, forks, surface and project area traits at the early stage of waterlogging stress; meanwhile, these root traits were reduced in waterlogging-sensitive genotypes (four representative genotypes) because of ABA and ethylene inhibition. Furthermore, waterlogging-tolerant barley genotypes, respectively, induced and inhibited the positive (*TCP20* and *PLT2*) and negative (*SHY2* and *PILS2*) regulated gene’s expressions that controlled the meristem growth at the root tips to cope with waterlogging stress. In addition, our results were consistent with the hypothesis that waterlogging-tolerant barley genotypes can simultaneously upregulate GA levels and downregulate ethylene accumulation, which may induce the expression of *SHR* and *SCR* genes and enhance root growth under waterlogging stress. This research lays a foundation for the subsequent screening and breeding of waterlogging-tolerant genotypes and the exploration of waterlogging-tolerant mechanisms in barley.

## 1. Introduction

Waterlogging stress is a major abiotic stress that affects agricultural development worldwide. Its impact not only limits the crop yields due to direct inundation, but also includes secondary disasters such as soil salinization, nutrient loss, and the outbreak of pests and diseases [1]. The United Nations Office for Disaster Risk Reduction (UNDRR) report (https://www.undrr.org/publication/undrr-annual-report-2021, accessed on 18 October 2025) indicated that in 2021, global agricultural economic losses caused by extreme precipitation reached 30 billion US dollars, with Asia being the most severely affected, accounting for 45%. Waterlogging leads to oxygen deficiency in the root system, affecting normal physiological functions of crops. Barley (*Hordeum vulgare* L.) is presently grown in over 100 countries across the globe. Nevertheless, there has been a notable reduction in global barley yields over the past two decades. A major driver of this yield decline is the escalating frequency of waterlogging events, combined with barley’s intrinsic susceptibility to the adverse impacts imposed by waterlogging stress [2]. Previous studies have documented that waterlogging stress can cause barley yield losses ranging from 40% to 79%, where such losses are contingent upon multiple factors including genotype, growth stage, soil properties, and duration of waterlogging exposure [3]. Therefore, screening waterlogging-tolerant genotypes helps stabilize crop yields under waterlogged conditions, minimize yield losses, ensure food supply, and meet the growing global food demand.

Modifications in root morphology play pivotal roles in plant adaptation to waterlogging stress, whereby plants develop adaptive traits such as aerenchyma formation-a process jointly regulated by ethylene and reactive oxygen species (ROS). Aerenchyma facilitates oxygen diffusion between above-ground plant tissues and submerged roots, thereby enhancing oxygen supply to the root system [4]. In barley, the proportion of aerated tissue in the root system of the waterlogging-tolerant genotype “Taixing 9425” reached 23.9% after waterlogging stress, which was significantly higher than that of the sensitive genotype “NasoNijo” [5]. Tolerant barley species, like *Hordeum marinum*, exhibit constitutive aerenchyma development, while cultivated genotypes often rely on ethylene-mediated induction of lysigenous aerenchyma under waterlogging [6]. In addition, plants also alter root growth directions and lengths to adapt to low-oxygen conditions. For instance, some plants exhibit reduced root tip elongation and increased lateral root growth, improving oxygen and nutrient absorption. In wheat, certain genes may regulate root cortical tissue changes through ACC (1-aminocyclopropane-1-carboxylic acid) or H_2_O_2_ signaling pathways [7,8,9,10]. Furthermore, root phenotypic plasticity (such as adventitious root proliferation) is also an important strategy. Waterlogging-tolerant barley genotypes expand the absorption surface area by increasing the density of lateral roots and the length of root hairs, thereby alleviating nutrient absorption disorders [11]. Additionally, waterlogging induces adventitious root growth and shallower root angles to escape hypoxic zones, though these responses vary among genotypes [12,13,14].

Hormone regulation is another critical aspect. Ethylene, abscisic acid (ABA), auxin, jasmonic acid, and gibberellins (GA) are all involved in waterlogging stress responses. Ethylene acts as a key signaling molecule promoting aerenchyma formation and adventitious root development [15]. ABA helps plants cope with waterlogging stress by regulating stomatal closure and reducing water loss [16]. Auxin affects root growth and development under waterlogged conditions by influencing cell elongation and division [17]. The latest research indicates that jasmonic acid and brassinolide are also involved in the regulation of waterlogging tolerance. For instance, brassinolide signaling maintains cellular ionic homeostasis by upregulating the activity of plasma membrane H^+^-ATPase, while jasmonic acid enhances resistance to secondary diseases by activating the expression of defense genes [18].

The development of transcriptomic and proteomic studies has revealed the molecular mechanisms of crop waterlogging tolerance. Under the stress of waterlogging, the hypoxic response genes (such as *ADH1*, alcohol dehydrogenase; *PDC*, pyruvate decarboxylase) in barley are rapidly activated, enhancing the anaerobic respiration capacity [11]. ADH1 encodes alcohol dehydrogenase, which catalyzes the conversion of pyruvate to ethanol and maintains a low level of ATP supply. PDC catalyzes the decarboxylation of pyruvate and reduces the accumulation of lactic acid [19]. The *TaBWPR-1.2* gene in wheat may regulate root cortical tissue changes through ACC or H_2_O_2_ signaling in response to waterlogging stress [7]. Additionally, overexpression of genes like *CIPK15* and *SnRK1* can improve sugar utilization efficiency under low-energy conditions or balance hormone responses, thereby promoting growth under waterlogging stress [20,21]. In conclusion, gramineous crops deploy diverse morphological, physiological, and molecular adaptations to waterlogging and hypoxia. Integrating these insights into breeding programs holds promise for enhancing resilience in flood-prone regions.

Waterlogging significantly reduces barley yield, with losses of 10–50% reported under field conditions, depending on the development stage of the waterlogged plant, duration of stress, temperature and type of soil [5,22,23]. Early-season waterlogging disrupts tillering and spike formation, while reproductive-stage stress impairs grain filling and increases lodging risk [24,25]. Above-ground parts show growth retardation, reduced plant height, fewer leaves, and smaller leaf area [11]. Photosynthesis and respiration are hindered, chlorophyll content drops, and leaf senescence accelerates. Waterlogging stress during the growing season affects the number of spikes, grains per spike, and thousand-grain weight. Studies have shown that barley yields are delayed maturity and increased lodging risk, exacerbated by weakened root systems [26]. Furthermore, waterlogging-induced soil redox changes can mobilize toxic ions (e.g., Fe^2+^, Mn^2+^) and reduce micronutrient availability, further compromising plant vigor [27,28].

In summary, waterlogging poses a multifaceted threat to barley production, necessitating interdisciplinary approaches that combine genetic improvement, agronomic management, and molecular insights to ensure food security in waterlogged regions. Despite the above reports, the molecular mechanisms underlying the regulation of hormonal metabolic pathways in adventitious roots in response to waterlogging are poorly understood in barley. Simultaneously, studies on waterlogging stress tolerance of different crop genotypes mainly focus on one or two growth stages. In this study, the main objectives were to analyze (i) the genotypic differences in waterlogging tolerance across whole growth stages to screen representative waterlogging-tolerant and waterlogging-sensitive barley genotypes; and (ii) the effects of the early stage waterlogging stress on root morphology, hormone content and relative gene expression to explore the mechanism of waterlogging stress tolerance in barley.

## 2. Results

### 2.1. Scanning Root Mophology Parameters After Hypoxic Stress at One-Leaf Stage

From the perspective of the whole growth period, this study takes 33 barley genotypes as research objects to explore the effects of waterlogging stress treatment at different stages (one-leaf stage, three-leaf stage, jointing stage, and booting stage) on barley. First, we investigated on the effect of 7 d hypoxic stress on root morphology by measuring root length, project area (projarea), surface area (surfarea), average diameter (avgdiam), tips and forks. These six physiological indicators measured were analyzed by relative changes which was calculated by (control-waterlogging)/control × 100%. The results showed that many genotypes had negative relative changes in all six root morphology indicators, especially in FPQ-6 and FPQ-8 (Figure 1), suggesting these genotypes had higher hypoxia-resistant abilities. In FPQ-8, the root length after hypoxia stress was increased to 2.9-fold compared to control (Figure 1A). In contrast, FPQ-7 and FPQ-26 were more sensitive to hypoxia stress compared to other genotypes (Figure 1). The number of root tips and forks were decreased to 53.04% and 54.85% in FPQ-7, respectively, while in FPQ-26, the number of root tips and forks decreased to 60.46% and 51.57% (Figure 1E,F).

### 2.2. Measuring Waterlogging Stress Effects During the Three-Leaf, Jointing, and Booting Stage

We further investigated the effects of three-leaf, jointing and booting stage waterlogging stress on different traits of 33 barley genotypes including relative chlorophyll content (SPAD), yellow/green leaf number, and plant height (Table 1). Waterlogging stress significantly reduced the values of different physiological indexes compared with control. Based on the mean values of SPAD, waterlogging stress treatments applied at three different growth stages all significantly affected the relative chlorophyll content, with the treatment at the three-leaf stage exerting the most pronounced impact compared with the other two stages (Table 1). Waterlogging stress imposed at the jointing stage exerted a significantly stronger impact on plant height and the number of green leaves compared to that applied at the booting stage (Table 1).

The seedlings processed in the above three stages of waterlogging treatments were continuously cultured normally until harvest. After harvest, the spike length and the number of kernels per spike of each genotype were detected. The genotypes with the greatest impact of waterlogging stress at different stages on spike length and kernels per spike were FPQ-7 and FPQ-13. Therein, waterlogging stress at jointing stage (JS) and booting stage (BS) led to the failure of producing grain in FPQ-7 and FPQ-13, respectively (Figure 2). Secondly, the spike length of genotype more significantly affected by waterlogging stress was FPQ-30, as it decreased to 62.53%, 67.70% and 72.38% compared to the control after three-leaf stage (TLS), JS, and BS waterlogging treatment (Figure 2A). Interestingly, we also found in FPQ-1 the spike length after waterlogging stress at TLS was increased to 2.04-fold compared to control but decreased to 4.54% and 20.41% at JS, and BS (Figure 2A). From the perspective of kernels per spike, FPQ-4 genotype had higher waterlogging tolerance ability compared to other genotypes which increased to 1.01-fold, 1.27-fold and 1.17-fold after TLS, JS, and BS waterlogging treatment, respectively (Figure 2B).

### 2.3. Evaluation of Waterlogging Tolerance in Different Barley Genotypes During the Whole Growth Period

By comprehensively integrating the waterlogging tolerance traits of root length, project area, surface area, average diameter, tips, forks, SPAD, green leaf number, plant height, spike length, kernels per spike, the average value of the waterlogging tolerance membership function values (MFVs) was calculated to evaluate the waterlogging tolerance of different genotypes during the whole growth period (Table 2). Comprehensive data analysis revealed that the MFV for 33 barley genotypes ranged from 0.19 to 0.76, with significant differences among the tested genotypes. Six genotypes with an average MFV smaller than 0.31 were classified as waterlogging-sensitive (WS) genotypes, four genotypes with an MFV more than 0.52 were classified as highly waterlogging-tolerant (HWT) genotypes, and the remaining genotypes were classified as moderate waterlogging-sensitive (MWS) and waterlogging-tolerant (WT) types.

### 2.4. Investigation on Root Traits of Waterlogging-Tolerant and -Sensitive Genotypes After Hypoxic Stress

We further investigated on the mechanisms of waterlogging tolerance between the waterlogging-sensitive genotypes (FPQ-2,11, 12, 13) and highly waterlogging-tolerant genotypes (FPQ-4, 6, 8, 24). Eight barley genotypes were treated with 7 d of hypoxic stress and root morphology was observed (Figure 3). After hypoxic stress, the root length and forks were significantly reduced in four waterlogging-sensitive genotypes compared to control; on the contrary, the root length and forks were increased in highly waterlogging-tolerant genotypes, especially in FPQ-8 genotype (Figure 3). Interestingly, we also found hypoxia stress increased the root surface area and project area in waterlogging-tolerant genotypes, except in FPQ-24 (Figure 4C,D).

### 2.5. Determining the Effects of Waterlogging Stress on Hormone Contents in Barley Roots

The GA, ABA, and ethylene precursor (ACC) contents in eight genotypes root after hypoxia stress were measured (Figure 5). The ABA content in four waterlogging-sensitive genotypes were significantly declined after hypoxia stress compared to control. Especially in FPQ-12 the ABA content was reduced to 55.89% relative to control (Figure 5A). Conversely, the ABA content was not significantly changed between waterlogging treatment and control in waterlogging-tolerant genotypes (Figure 5A). The GA and ACC content did not show the unified trend of changes similar to ABA content after hypoxia stress (Figure 5B,C). Generally, the ACC content was significantly decreased after hypoxia stress in waterlogging-sensitive genotypes except FPQ-2, while there were no significant changes in waterlogging-tolerant genotypes except FPQ-4. In four waterlogging-sensitive genotypes, hypoxia stress did not significantly affect the GA content (Figure 5C). However, among the four waterlogging-tolerant genotypes, the content of GA in two of them significantly increased (Figure 5C).

### 2.6. Detecting the Effects of Waterlogging Stress on Root Development-Related Genes in Barley

According to previous reports, six genes related to root development were selected to detect their expression level changes in the roots of eight barley genotypes. Unfortunately, we did not find the unified gene expression pattern between waterlogging-sensitive and waterlogging-tolerant genotypes (Figure 6) due to the sophisticated regulation networks in root. Even so, we found the expressions of *TCP20* (teosinte-branched cycloidea PCNA factor 20) were all decreased in waterlogging-sensitive genotypes which was similar on *SCR* (SCARECROW) gene except in FPQ-13 (Figure 6A,F). The expressions of *SHY2* (SHORT HYPCOTYL 2) were significantly increased in waterlogging-sensitive genotypes except FPQ-13, while this gene was decreased in waterlogging-tolerant genotypes except FPQ-4 (Figure 6D). The expressions of *PLT2* (PLETHORA 2) were significantly declined in waterlogging-sensitive genotypes except FPQ-2,12, while this gene was increased in waterlogging-tolerant genotypes except FPQ-6,8 (Figure 6B). The expression of *SHR* (SHORT-ROOT), *PILS2* (PIN-LIKES 2) did not show genotype specificity that induced and declined in both waterlogging-sensitive and waterlogging-tolerant genotypes (Figure 6C, E).

## 3. Discussion

Plants differ greatly in their tolerance of waterlogging, as reflected in their different growth responses. Most plants have developed waterlogging-resistant mechanisms, but their abilities varies widely among different species and cultivars. Similarly to other abiotic stresses, waterlogging tolerance is a complicated trait and the selection for waterlogging-tolerant varieties has been a major obstacle in crop breeding. This study provides a comprehensive evaluation of the progressive effects of waterlogging at four different growth stages (one-leaf stage, three-leaf stage, jointing stage, and booting stage) on barley through an integrated analysis of root morphology, chlorophyll content, yellow/green leaf number, plant height, spike length and kernels per spike in different barley genotypes.

In the early stage of waterlogging stress, plant roots are the first to perceive hypoxic stress. In this study, the changes in root morphology of barley at the one-leaf stage were mainly investigated. There are significant phenotypic differences between waterlogging-tolerant genotypes and waterlogging-sensitive genotypes among the 33 barley varieties. For example, the root length of the waterlogging-tolerant genotype FPQ-8 was increased by 1.88-fold after hypoxia stress; meanwhile, the waterlogging-sensitive genotype FPQ-7 was decreased by 42.79% compared to control (Figure 1). Waterlogging leads to a drastic reduction in root length, biomass and root-to-shoot ratio in barley [29,30] and wheat [31,32]. However, in our study we found the six root morphology traits including root length, project area, surface area, average diameter, tips and forks of some waterlogging-tolerant genotypes such as FPQ-4, 6, 8 were all increased after 7 days of hypoxic stress compared to control (Figure 1). These observations are also in line with another study which reported that the total adventitious root length, number and surface area is increased relative to control after waterlogging treatment in waterlogging-tolerant barley genotypes [33]. It is suggested that waterlogging-tolerant genotypes increases root length, project area, surface area average diameter, tips and forks traits to response to early stage of waterlogging stress. Several studies have demonstrated that these root traits are controlled by plant hormones. We will conduct a detailed discussion in the following paragraphs.

The tolerance abilities to waterlogging stress during the different crop ontogeny significantly varies at different growth stages with different phenotypes presented. Leaf chlorosis was the main criteria at the early stage of waterlogging [34]. In our study we measured the yellow leaf number at the three-leaf stage. During the later stage of development (at the jointing stage and booting stage), due to physiological aging, there were too many yellow leaves. Therefore, we began to investigate the number of green leaves. At early stages of waterlogging stress, susceptible genotypes showed much faster reaction to waterlogging stress with more than one leaf turning yellow while the tolerant genotypes only showed a slight symptom on the tip of the first leaf (Table 1), which was also demonstrated in other studies [5,34]. Previous studies have reported that the susceptibility of barley to waterlogging stress varies across different growth stages, with the order of sensitivity being grain filling stage > tillering stage > seedling stage [2]. This was also proved in our study that the waterlogging effected the most damage at the booting stages compared to the three-leaf stage and jointing stage from the spike length and kernels per spike parameters (Table 1), since waterlogging stress at the early stages allowed for an effective recovery of physiological and growth parameters by the time of the critical period for yield determination.

The genotype presents notable differences in waterlogging tolerance. The average value of the waterlogging tolerance MFV was calculated to evaluate the waterlogging tolerance of different genotypes during the whole growth period and the most four sensitive and four highly tolerant genotypes were chosen for the further study. The roots of the plants are the first to be affected by waterlogging stress. Some species present variable ability to generate adventitious roots with aerenchyma to facilitate the diffusion of oxygen from the shoot into and along the roots [35,36]. Our previous research showed that the number of differentially expressed genes in the roots of barley under waterlogging stress was significantly higher than that in leaves [37]. Therefore, we have decided to focus on the root system as the main subject for the study of barley’s mechanism of tolerance to waterlogging stress. In the present study, waterlogging stress increased the root total length, forks, surface area and project area in waterlogging tolerance genotypes while decreased in waterlogging-sensitive genotypes (Figure 3 and Figure 4) which proved that barley enhanced the root system response to waterlogging stress.

Waterlogging affects the root growth which has been demonstrated to be controlled by plant hormones. Previous reports prove both epidermal programmed cell death and adventitious root growth are regulated through the interaction of ABA, ethylene and GA [38,39]. In the present study, the ABA, ACC (ethylene precursor) and bioactive GA were measured in roots of eight representative genotypes (Figure 5). In waterlogging-sensitive genotypes of barley, the ABA accumulation and ACC content were significantly decreased in waterlogging-sensitive genotypes except FPQ-2, while the GA content did not significantly change after waterlogging stress (Figure 5). Based on these results, we suggested that the inhibition of root growth in waterlogging-sensitive genotypes under waterlogging conditions mainly relied on the ABA and ethylene. However, in other plants such as Arabidopsis and wheat the root ABA level is not affected by long-term flooding/oxygen-deficient conditions [14,40], which is consistent with the results we found in waterlogging-tolerant genotypes. Therefore, we infer that barley with varying waterlogging resistance abilities may respond to waterlogging stress through different hormone regulation mechanisms.

The *PLT* gene is required for root stem cell specification and maintenance. This gene was proven genetically and physically interact with plant-specific *TCP20* transcription factors to specify the stem cell niche during embryogenesis and maintain organizer cells post-embryonically [41]. We found the expression levels of these two positively regulated genes were both decreased in the waterlogging-sensitive genotypes except in FPQ-2,12, and the expression levels of these two genes increased in waterlogging-tolerant genotypes FPQ-6 and 8 (Figure 6A,B). At the same time, we also found the expression of *SHY2* and *PILS2*, both as negative regulators in root meristem development, [42] were declined after waterlogging stress in waterlogging-tolerant genotypes except in FPQ-4 and 6 (Figure 6D,E). Based on these results, we proposed a hypothesis that most waterlogging-tolerant barley genotypes may, respectively, induce and inhibit the positive and negative regulated gene expressions that control the meristem at the root tips to cope with waterlogging stress. The further functional validation experiment is needed and explored in future research.

In root, GA in combination with the SHR/SCR module additively or synergistically act in the control of middle cortex formation [43]. Here we found the GA content was increased in waterlogging tolerance genotype (FPQ-8) after waterlogging stress; in the meantime, the expression of *SHR* and *SCR* was also increased (Figure 5 and Figure 6). Furthermore, ethylene downregulates the expression of *SHR* and *SCR* genes encoding other growth-stimulating proteins [44]. In this study, the ACC content was decreased in waterlogging tolerance genotype (FPQ-4) and the expression of *SHR* and *SCR* were all increased after waterlogging stress (Figure 5 and Figure 6). Based on these findings, our results were consistent with the hypothesis that waterlogging-tolerant barley genotypes can simultaneously upregulate GA levels and downregulate ethylene accumulation, which may induce the expression of *SHR* and *SCR* genes and enhance root growth under waterlogging stress. Among the waterlogging-sensitive genotypes, a notable reduction in ABA content was detected (Figure 5A). Previous studies have indicated that ABA suppresses the expression of the *SHY2* [45]. In the present study, it was observed that with the exception of the FPQ-13 genotype, the expression levels of the *SHY2* in the other three waterlogging-sensitive genotypes were upregulated following waterlogging stress treatment (Figure 6D). Given that the SHY2 plays a negative regulatory role in root meristems [42], we speculate that the inhibitory effect of waterlogging stress on the root growth of waterlogging-sensitive genotypes is associated with the regulation of the *SHY2* gene by ABA, which needs to perform further validation in barley mutants.

## 4. Materials and Methods

### 4.1. Materials and Root Morphology Scanning

In addition to NasoNijo and Fergus, two genotypes from Japan and Canada, respectively, 31 barley genotypes from different regions within China were selected, making a total of 33 genotypes for the waterlogging stress treatment experiment (Appendix A). When the 33 barley genotypes were germinated and cultured for one week using the rolled paper method under the conditions of 22 °C, 18 h light, and 6 h darkness, they were transferred to half-strength Hoagland’s nutrient solution. For the control group, the nutrient solution was continuously aerated to ensure aerobic conditions for the seedlings. In contrast, the seedlings subjected to waterlogging treatment were not aerated. After 7 d of treatment, the root morphology including root length, project area (projarea), surface area (surfarea), average diameter (avgdiam), tips and forks was scanned using a multi-purpose root system analysis system, and the data were processed with WinRHIZO Pro 2012b software. These physiological indicators were analyzed by relative changes ((control-waterlogging)/control × 100%).

### 4.2. Waterlogging Treatments

This experiment was conducted in a glass greenhouse at Yangzhou University. For every three genotypes, 6 seeds of each genotype were sown in the same pot (25 × 30 × 25 cm), which was filled with a substrate and soil mixture at a 1:1 ratio. Each genotype had 8 replicate flowerpots. Specifically, 2 replicates served as the control group, 2 replicates were subjected to waterlogging treatment at the three-leaf stage, 2 replicates at the jointing stage, and 2 replicates at the booting stage. When the seedlings reached the three-leaf stage, the flowerpots of 2 replicates were placed in a pool for waterlogging treatment. The water level was maintained at 1 cm above the soil surface for 2 weeks. After the treatment, relative chlorophyll content (SPAD) and the number of yellow leaves were measured. When the seedlings reached the jointing stage, the flowerpots of another 2 replicates were placed in the pool for waterlogging treatment, with the water level kept 1 cm above the soil surface for 2 weeks. Upon completion, SPAD, plant height, and the number of green leaves were determined. When the seedlings reached the heading stage, the flowerpots of the remaining 2 replicates were subjected to waterlogging treatment in the pool, maintaining the water level 1 cm above the soil surface for 2 weeks. After treatment, SPAD, plant height, and the number of green leaves were measured. Additionally, after all treatments, the seedlings continued to grow under normal watering conditions until seed harvest. Before seed harvest, the kernels per spike and spike length were measured. During the measurement, 12 biological replicates were used for all the parameters including SPAD, plant height, the number of yellow or green leaves, the kernels per spike and spike length measurements per genotype and treatment.

### 4.3. Evaluation of Waterlogging Tolerance

The membership function value (MFV) of waterlogging tolerance was applied as a comprehensive index to evaluate the waterlogging tolerance of 33 barley genotypes according to the related 11 traits. These traits included root length, project area (projarea), surface area (surfarea), average diameter (avgdiam), tips and forks at one-leaf stage; SPAD, green leaf number, plant height, spike length, kernels per spike at three-leaf stage (TLS), jointing stage (JS) and booting stage (BS). The MFV was estimated via the following equation:U_ij_ = (D_ij_ − D_jmin_)/(D_jmax_ − D_jmin_)
$$U_{i}=\frac{1}{n}\sum_{j}^{n}U_{ij}$$

where D was calculated as the ratio of the value for the waterlogged and the control for all the traits; U_ij_ represents the MFV of the j trait for the i barley genotype; D_ij_ represents the actual measured value of the j trait in the i genotype; D_jmax_ and D_jmin_ are the maximum and minimum values of the j trait observed in all barley genotypes, respectively; and U_i_ is the mean value of the MFV for all traits for the i barley genotypes.

### 4.4. Hormone Contents Measurements

From the above 33 barley genotypes, four waterlogging-tolerant genotypes and four waterlogging-sensitive genotypes were selected. They were cultured to one-week-old using the rolled paper method, followed by one-week hypoxia stress treatment (using the same method as above). Thereafter, their roots were collected for the determination of GA, ABA, and ethylene precursor contents. The hormone contents were measured by Mlbio reagent kit (YJ-077235, 455012, 906901) according to their instructions. Briefly, after the stress treatment, root tissues were collected, weighed, and immediately preserved in liquid nitrogen. For detection, tissue extraction solution was added to the preserved samples, followed by grinding. The homogenate was centrifuged at 2000 rpm for 20 min, and the supernatant was collected for subsequent assays using commercial kits. A standard curve was generated with the standard substances provided in the kit, and the hormone content of the experimental samples was calculated based on their detected values, with the unit expressed as pmol/L.

### 4.5. Real-Time PCR

The above eight barley genotypes were cultured to one-week-old, subjected to one-week waterlogging stress treatment (using the same method as above), and then their roots were collected and rapidly frozen in liquid nitrogen. The total RNA in root was extracted and real-time PCR was performed with the ChamQ SYBR qPCR Master Mix kit using a CFX96 thermocycler (Bio-Rad, Hercules, CA, USA). The PCR program had two steps: one cycle of 95 °C, 30 s; 40 cycles of 95 °C, 10 s; 60 °C, 30 s. Three biological and two technical replicates were performed for each treatment. The root growth relative genes including teosinte-branched cycloidea PCNA factor 20 (*TCP20*), plethora 2 (*PLT2*), PIN-LIKES 2 (*PILS2*), SHORT-ROOT (*SHR*), SCARECROW (*SCR*), SHORT HYPCOTYL 2 (*SHY2*) were quantified with gene specific primers (Appendix A). Ubiquitin (*UBI*) was used as the reference gene.

### 4.6. Statistical Analysis

The SPSS 21 software was used for data analysis. Data presented are the mean values ± SE. A significance of the difference between multiple samples were estimated by one-way ANOVA followed by Tukey’s test. Means with different letters indicate statistically significant difference (*p* < 0.05). The significance levels are * *p* < 0.05, ** *p* < 0.01, and *** *p* < 0.001.

## 5. Conclusions

Waterlogging tolerance is a complex trait affected by several mechanisms and complicated by confounding factors. At present, studies on waterlogging stress tolerance of different crop genotypes mainly focus on one or two specific stages. In this study, 33 barley genotypes were taken as the research objects and the phenotypes during the whole growth stages were investigated to comprehensively assess their waterlogging tolerance abilities. Then, we chose the most waterlogging-sensitive and -tolerant genotypes as representative objects and measured their root morphology, hormone content and relative root development gene expressions to explore the mechanism of waterlogging stress tolerance in barley. In summary, we find the waterlogging-tolerant genotypes response to waterlogging stress through increasing root length, forks, surface and project area traits at the early stage of waterlogging stress. In contrast, these root traits are declined in waterlogging-sensitive genotypes which is probably dependent on the ABA and ethylene regulation as ABA presumably induces the expression of the *SHY2* to inhibit the root growth. Further exploration reveals that the *TCP20* and *PLT2* genes which positively control the meristem growth at the root tip are induced while the negative regulated genes (*SHY2* and *PILS2*) are inhibited in most waterlogging-tolerant genotypes which explains why the waterlogging-tolerant genotypes could increase root length traits under waterlogging stress. We also find the waterlogging tolerance barley genotypes concurrently elevate the levels of GA and reduce the ethylene, inducing the expression of *SHR* and *SCR* genes and possibly promoting root growth under waterlogging stress. However, these findings still need further functional validation experiments in barley.

## Figures and Tables

**Figure 1 plants-15-00134-f001:**
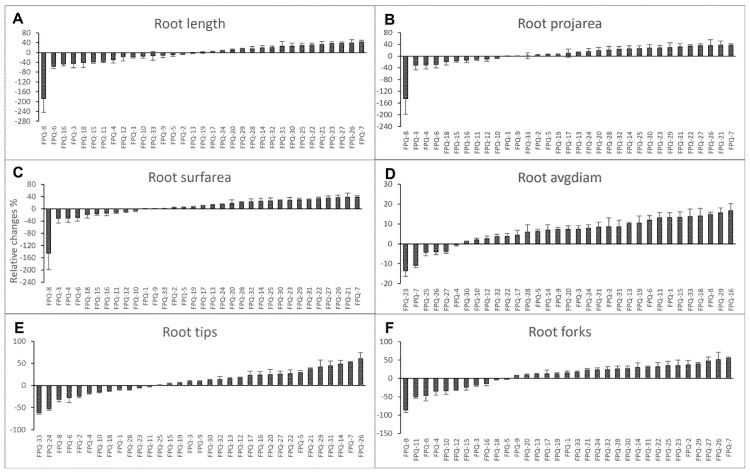
The relative changes in root length (**A**), project area (projarea) (**B**), surface area (surfarea) (**C**), average diameter (avgdiam) (**D**), tips (**E**) and forks (**F**) in 33 barley genotypes after one week of hypoxic stress. The relative changes were calculated by (control-waterlogging)/control × 100%. Data are the mean ± SE.

**Figure 2 plants-15-00134-f002:**
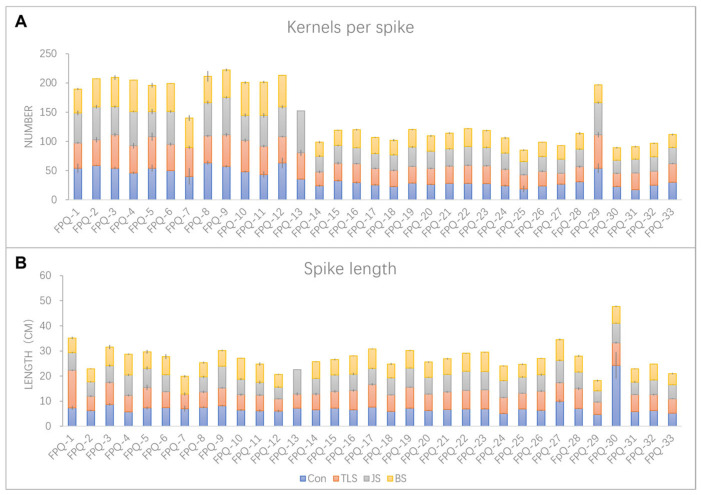
The spike length (**A**) and kernels per spike (**B**) in 33 barley genotypes under control (Con), three-leaf stage (TLS) waterlogging, jointing stage (JS) waterlogging and booting stage (BS) waterlogging treatment. Data are the mean ± SE.

**Figure 3 plants-15-00134-f003:**
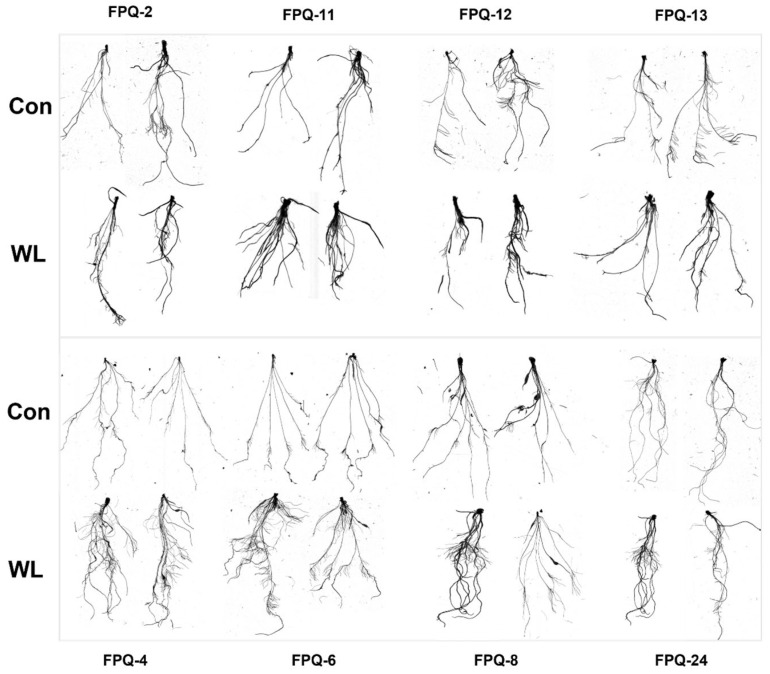
The image of roots in 8 barley genotypes under control (Con) and waterlogging (WL) condition. The top panel image represents the waterlogging-sensitive genotypes while the bottom panel image represents the waterlogging-tolerant genotypes.

**Figure 4 plants-15-00134-f004:**
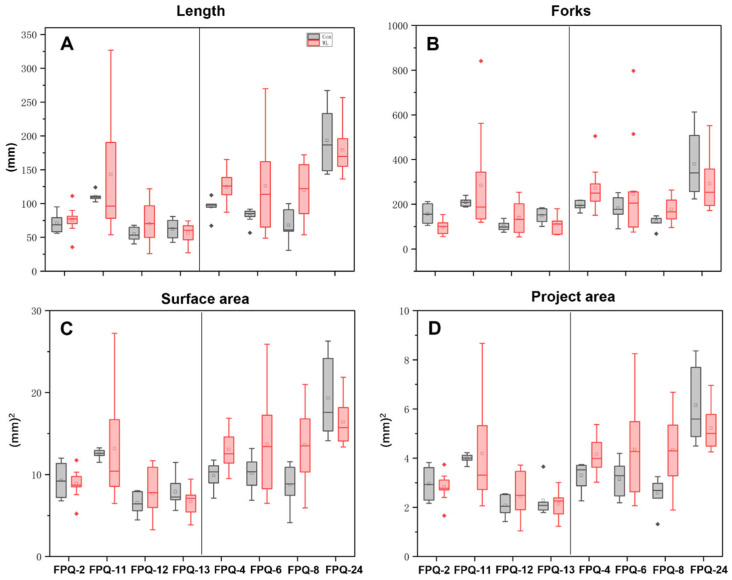
Box plot of root length (**A**), root forks (**B**), root surface area (**C**), root project area (**D**), in 8 barley genotypes after hypoxia treatment. Square represents mean value; line in the box indicates median value; bars represent the first and third quartile. Red and gray rhombuses mark outliers. In each four panels, the left part represents the waterlogging-sensitive genotypes while the right part represents the waterlogging-tolerant genotypes.

**Figure 5 plants-15-00134-f005:**
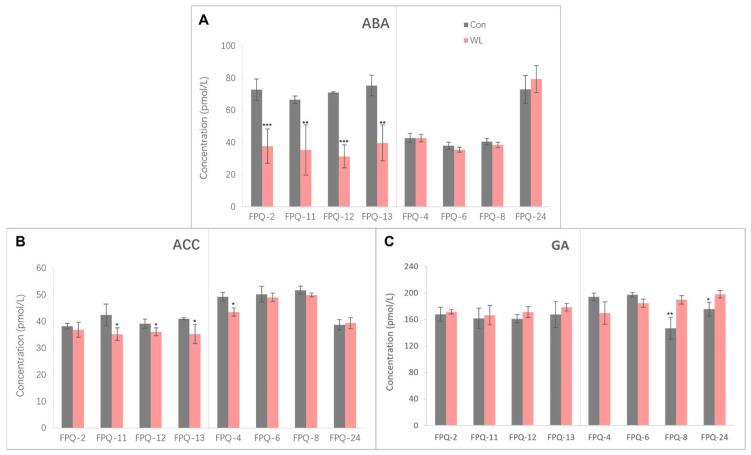
The abscisic acid (ABA) (**A**), ethylene precursor (ACC) (**B**), gibberellin (GA) (**C**) contents in eight genotypes’ roots. The gray bars represent control (Con), and the red bars represent waterlogging (WL) treatment. The significance levels are * *p* < 0.05, ** *p* < 0.01, and *** *p* < 0.001. In each panel, the left part represents the waterlogging-sensitive genotypes while the right part represents the waterlogging-tolerant genotypes.

**Figure 6 plants-15-00134-f006:**
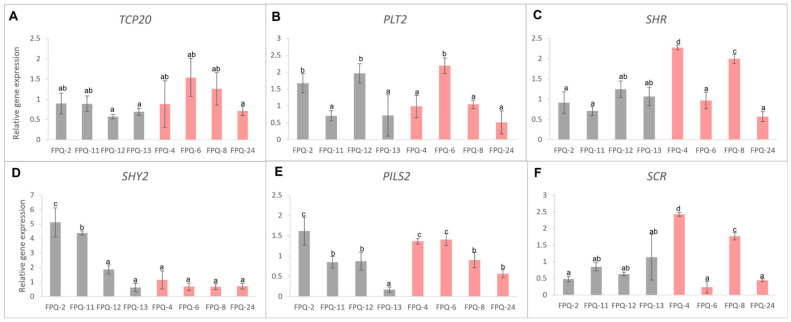
The relative gene expressions inlcuding (**A**) *TCP20*, (**B**) *PLT2*, (**C**) *SHR*, (**D**) *SHY2*, (**E**) *PILS2* and (**F**) *SCR* in eight barley genotypes’ roots. Different lowercase letters represent significant differences between different varieties at *p* < 0.05. Gray bars represent waterlogging-sensitive genotypes and red bars represent waterlogging-tolerant genotypes.

**Table 1 plants-15-00134-t001:** The descriptive statistics of waterlogging tolerance related traits in 33 barley genotypes after different waterlogging treatments during three stages.

	Three-Leaf Stage			Jointing Stage						Booting Stage					
	SPAD		Yellow Leaf No.	SPAD		Green Leaf No.		Plant Height		SPAD		Green Leaf No.		Plant Height	
	Con	WL	WL	Con	WL	Con	WL	Con	WL	Con	WL	Con	WL	Con	WL
**Mean**	46.08	30.25	0.56	55.23	46.92	52.18	31.22	58.51	47.42	51.39	43.91	26.79	23.42	68.39	64.43
**Maximum**	51.20	36.46	1.51	62.30	54.60	77.00	59.57	78.00	60.86	60.28	58.35	42.67	37.20	88.75	82.80
**Minimum**	41.50	23.66	0.08	44.86	39.38	20.67	16.00	39.08	35.70	35.07	25.18	11.50	12.30	42.00	42.88
**CV%**	5.14	10.47	64.72	7.07	8.72	23.02	28.27	15.67	15.74	9.92	16.81	23.45	29.30	16.87	16.81
**Treatment**	***		-	***		***		***		***		*		*	
**Cultivar**	***		***	***		***		***		***		***		***	
**T × C**	***		-	*		***		*		NS		*		NS	

Note: SPAD: soil and plant analyzer development, relative chlorophyl content; Con: control; WL: waterlogging; no.: number; CV: variation coefficients; T × C: treatment vs. control. The significance levels are * *p* < 0.05 and *** *p* < 0.001.

**Table 2 plants-15-00134-t002:** Membership function values (MFVs) of highly waterlogging-tolerant (HWT), waterlogging-tolerant (WT), moderate waterlogging-sensitive (MWS) and waterlogging-sensitive (WS) barley germplasms during the complete reproductive stages.

Genotype	MFV	Standard Grade	Genotype	MFV	Standard Grade
FPQ-12	0.19	WS	FPQ-3	0.38	MWS
FPQ-13	0.22		FPQ-22	0.39	
FPQ-2	0.25		FPQ-20	0.39	
FPQ-11	0.27		FPQ-16	0.39	
FPQ-29	0.30		FPQ-33	0.40	
FPQ-30	0.30		FPQ-19	0.40	
FPQ-27	0.31	MWS	FPQ-10	0.40	
FPQ-15	0.34		FPQ-17	0.41	WT
FPQ-5	0.34		FPQ-23	0.44	
FPQ-21	0.35		FPQ-31	0.44	
FPQ-7	0.35		FPQ-18	0.46	
FPQ-26	0.35		FPQ-25	0.52	
FPQ-28	0.35		FPQ-6	0.55	HWT
FPQ-1	0.36		FPQ-8	0.62	
FPQ-14	0.37		FPQ-4	0.73	
FPQ-32	0.37		FPQ-24	0.76	
FPQ-9	0.38				

## Data Availability

All data can be found in the main text and the Appendix A.

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
