# Peer review of "Strong Root System Enhances Waterlogging Resilience in Barley (Hordeum vulgare) at the Early Stage Stress"

_plants, 2026, doi:10.3390/plants15010134_

Round 1
Reviewer 1 Report
Comments and Suggestions for Authors
- The introduction needs to be condensed because it contains excessive repetitive content. At the same time, the authors must clearly explain the key differences between the research objectives of this manuscript and those of previously published literature. Although this has been mentioned, it lacks support from literature-based argumentation. In other words, the length and focus of the authors’ discussion are misplaced, and overall adjustments are needed.
- The phenotypic data need to include statistical information such as standard deviation.
- For the same genotype, the phenotypic statistical data under waterlogging and non-waterlogging conditions should be compared with each other.
- The results indicate that only 4 waterlogging-tolerant and 4 waterlogging-sensitive varieties were examined at the genotype level. This shows that not all varieties were investigated. The authors need to clearly explain this in the abstract, introduction, and methods sections, especially in the discussion and conclusion.
- The content of this study is interesting, but both the title and the discussion lack a writing framework that aligns the experiment with the data. Without solid statistical analyses, many of the arguments remain one-sided inferences. In addition, although the title emphasizes a specific developmental stage, the content does not seem to match the title. Beyond the simple issues listed above, the overall writing quality of the manuscript is poor. The authors must further examine the data, reorganize the logical flow, and rewrite the manuscript to verify the correctness and reliability of the study. Without proper argumentation, this manuscript will not meet the basic requirements of a scientific journal.
- There is substantial textual overlap between this manuscript and previously published articles. The authors must be extremely careful regarding academic ethics.
- The root-related data appear to be inferred merely from pixel values, yet the manuscript relies heavily on this information to determine whether the genotypes are waterlogging-tolerant. The authors must be extremely careful, as many morphological changes cannot be accurately judged from 2D images alone.
Author Response
Author Response to Reviewer Comments
- Comment:The introduction needs to be condensed because it contains excessive repetitive content. At the same time, the authors must clearly explain the key differences between the research objectives of this manuscript and those of previously published literature. Although this has been mentioned, it lacks support from literature-based argumentation. In other words, the length and focus of the authors’ discussion are misplaced, and overall adjustments are needed.
Reply: Thanks for your comment. The introduction has been condensed and highlighed in red. The discussion was also revised as you suggested.
- Comment:The phenotypic data need to include statistical information such as standard deviation.
Reply: Thanks for your comment. The statistical information has been added.
- Comment:For the same genotype, the phenotypic statistical data under waterlogging and non-waterlogging conditions should be compared with each other.
Reply: Thanks for your comment. For Fig1, it showed the relative changes after waterlogging stress which was calculated as (control-waterlogging)/control*100%; for Fig 2, it showed the spike length and kernels per spike in different genotypes after waterlogging stress at three-leaf stage (TLS), jointing stage (JS) and booting stage (BS); for Fig 3, it showed the root traits of the most sensitive and tolerant genotypes under control and waterlogging stress; for Fig 4, it showed the root length, forks, surface area and project area of the most sensitive and tolerant genotypes under control and waterlogging stress.
- Comment:The results indicate that only 4 waterlogging-tolerant and 4 waterlogging-sensitive varieties were examined at the genotype level. This shows that not all varieties were investigated. The authors need to clearly explain this in the abstract, introduction, and methods sections, especially in the discussion and conclusion.
Reply: Thanks for your comment. In the whole text these 8 genotpes had been explained and representatively highlighed in red.
- Comment:The content of this study is interesting, but both the title and the discussion lack a writing framework that aligns the experiment with the data. Without solid statistical analyses, many of the arguments remain one-sided inferences. In addition, although the title emphasizes a specific developmental stage, the content does not seem to match the title. Beyond the simple issues listed above, the overall writing quality of the manuscript is poor. The authors must further examine the data, reorganize the logical flow, and rewrite the manuscript to verify the correctness and reliability of the study. Without proper argumentation, this manuscript will not meet the basic requirements of a scientific journal.
Reply: Thanks for your comment. Following your suggestions, the authors have carefully revised the Discussion section, and conducted statistical analyses for Figure 1 and Figure 2 respectively.
The overall design rationale of this study is as follows: Barley cultivars collected from major barley-producing regions in China were subjected to waterlogging stress treatments across multiple growth stages— a notable improvement over previous studies, which mostly focused on only one or two stress stages. Through comprehensive evaluation, 8 representative cultivars were selected, including both waterlogging-tolerant and waterlogging-sensitive genotypes.
Subsequent root phenotypic analysis revealed that the root systems of waterlogging-tolerant barley cultivars became even more robust after waterlogging stress, whereas those of waterlogging-sensitive cultivars were severely impaired (this finding has also been reported in other studies; Luan et al; 2023). To elucidate the underlying mechanisms, we further analyzed the changes in hormone contents and the expression levels of genes related to adventitious root formation. The results indicated that the enhanced root vigor of waterlogging-tolerant barley cultivars under waterlogging stress induced and inhibited the positive (TCP20 and PLT2) and negative (SHY2 and PILS2) regulated gene’s expressions that controlled the meristem growth at the root tips to cope with waterlogging stress. Besides, waterlogging tolerance genotypes concurrently elevated the levels of GA and reduced the ethylene, thereby inducing the expression of SHR and SCR and promoting root growth in barley.
Currently, numerous studies have investigated the effects of waterlogging stress on plant adventitious root growth. However, most previous research has only focused on identifying certain genes that may be involved in regulating adventitious root formation, or relied on visual observation to assess the number of adventitious roots during phenotypic identification (Manik et al; 2022 ; Mano et al; 2005).
For the first time, this study provides a clear and accurate determination of the effects of waterlogging stress on barley adventitious root growth, and further conducts an in-depth analysis of the underlying mechanisms. Therefore, this research is of great significance for elucidating the regulatory mechanisms of plant tolerance to waterlogging stress.
- Luan H, Li H, Li Y, Chen C, Li S, Wang Y, Yang J, Xu M, Shen H, Qiao H. 2023. Transcriptome analysis of barley (Hordeum vulgare L.) under waterlogging stress, and overexpression of the HvADH4 gene confers waterlogging tolerance in transgenic Arabidopsis. BMC Plant Biol 23, 1-22.
- Manik SMN, Quamruzzaman M, Livermore M, Zhao CC, Johnson P, Hunt I, Shabala S, Zhou MX. 2022. Impacts of barley root cortical aerenchyma on growth, physiology, yield components, and grain quality under field waterlogging conditions. Field Crops Research 279.
- Mano Y, Muraki M, Fujimori M, Takamizo T, Kindiger B. 2005a. Identification of QTL controlling adventitious root formation during flooding conditions in teosinte (Zea mays ssp.
Huehuetenangensis) seedlings. Euphytica, 142, 33–42.
- Comment:There is substantial textual overlap between this manuscript and previously published articles. The authors must be extremely careful regarding academic ethics.
Reply: Thanks for your comment. The manuscript has been revised carefully.
- Comment:The root-related data appear to be inferred merely from pixel values, yet the manuscript relies heavily on this information to determine whether the genotypes are waterlogging-tolerant. The authors must be extremely careful, as many morphological changes cannot be accurately judged from 2D images alone.
Reply: Thanks for your comment. As replyied in Comment 5, the waterlogging-tolerant or -sensitive barley cultivars was evaluated by MFV which was calculated by 11 traits (root length, project area, surface area, average diameter, tips, forks, SPAD, green leaf number, plant height, spike length, kernels per spike) during four stages of waterlogging stress in 33 barley genotypes (more details in line 191-202). Then, we further analysed the root mophology in the 8 genotypes which indeed based on the pixel values. However, beside the pixel values, we also had the phynotype images which chould clearly show the different root traits between waterlogging-tolerant and -sensitive barley cultivars.
Reviewer 2 Report
Comments and Suggestions for Authors
Review of the article «Strong Root System Achieves Waterlogging Resilience in Barley (Hordeum vulgare L.) at the Early Stage of Waterlogging Stress» by Yu Tian, Li Cao, Yuening Xin, Liang Zhu, Zhenxiang Zhou, Baojian Guo, Chao Lv, Juan Zhu, Rugen Xu, Feifei Wang Jiangsu
- The title of the article and the Latin spelling of barley should be replaced from «Hordeum vulgare. L» to «Hordeum vulgare». Furthermore, the title contains stylistic redundancies. For example, it contains the word "waterlogging" twice. I believe this should be avoided. The verb «achieves» sounds too categorical. It suggests that a strong root system is the main mechanism of resistance. However, these articles (the influence of hormones, gene expression) show a correlation, but do not 100% prove a direct cause-and-effect relationship.
- Sentences need to be more precisely worded. For example, lines 20-21 state in a general phrase, «For now, little is known about the molecular mechanisms underlying the regulation of hormonal pathways in barley roots». Considering that the previous sentence refers to several stress factors, it is unclear which of them the phrase about molecular mechanisms referred to.
- Section titles. I believe they should be written more clearly. In your article, the headings look like spoilers. Also, in my opinion, the title of section 2.1 does not reflect its content. If this section concerns the analysis of data at the seedling stage, that is how it should be written. Similar comments apply to 2.2, 2.4-2.6. Section titles should not include secondary information, such as the number of genotypes. Furthermore, the first sentence in section 2.1 does not apply only to 2.1. It is an introductory phrase applicable to all sections.
- Coefficient of Variation Analysis. This is a basic statistical parameter that indicates the level of variability. It is generally accepted that a CV of up to 10% indicates low variability, 11-20% indicates moderate variability, and above 20% indicates high variability. Numerous studies have also shown that under stress conditions, variability increases. This is primarily due to sample heterogeneity, that is, the different ability of plants within a single genotype to respond to a stress factor. Why did the authors choose to use this parameter for discussion in the text of the article? It is usually not given such close attention. It is more appropriate to analyze mean or median trait values in detail, rather than CV.
- A clarification about the dots should be added to the caption for Figure 4 – are they outliers or extremes?
- The missing «*», «**», and «***» in Figure 5 should be added, as the captions are present.
- It is unclear what is meant by GA3? The article lacks a description of one of the GA types (GA3) studied in the work. The list of abbreviations only includes GA.
- In all figures where 8 genotypes are indicated, it is necessary to indicate which of them are sensitive and which are resistant. This will facilitate the understanding of the material.
- In the introduction and discussion, the authors discuss the importance of aerenchyma formation under hypoxic conditions, but they do not study this process themselves. How is aerenchyma formation related to salicylic acid synthesis, especially during the initial stages of plant development? Also of interest in studying aerenchyma formation is the analysis of the expression of the β2 proteasome subunit (PBB gene) under hypoxic conditions.
- The authors conclude that root growth inhibition in sensitive genotypes depends on ABA and ethylene, i.e., only on the parameters they studied. Data showed that ABA levels decreased in these genotypes. However, the classic role of ABA as a stress hormone would suggest an increase.
Author Response
Author Response to Reviewer Comments
- Comment: The title of the article and the Latin spelling of barley should be replaced from «Hordeum vulgare. L» to «Hordeum vulgare». Furthermore, the title contains stylistic redundancies. For example, it contains the word "waterlogging" twice. I believe this should be avoided. The verb «achieves» sounds too categorical. It suggests that a strong root system is the main mechanism of resistance. However, these articles (the influence of hormones, gene expression) show a correlation, but do not 100% prove a direct cause-and-effect relationship.
Reply: Thanks for your comment. The title has been chaged to “Strong root system enhances waterlogging resilience in barley (Hordeum vulgare) at the early-stage stress”.
- Comment: Sentences need to be more precisely worded. For example, lines 20-21 state in a general phrase, «For now, little is known about the molecular mechanisms underlying the regulation of hormonal pathways in barley roots». Considering that the previous sentence refers to several stress factors, it is unclear which of them the phrase about molecular mechanisms referred to.
Reply: Thanks for your comment. This sentence has been changed to “For now, little is known in waterlogging stress about the molecular mechanisms underlying the regulation of hormonal pathways in barley roots”.
- Comment: Section titles. I believe they should be written more clearly. In your article, the headings look like spoilers. Also, in my opinion, the title of section 2.1 does not reflect its content. If this section concerns the analysis of data at the seedling stage, that is how it should be written. Similar comments apply to 2.2, 2.4-2.6. Section titles should not include secondary information, such as the number of genotypes. Furthermore, the first sentence in section 2.1 does not apply only to 2.1. It is an introductory phrase applicable to all sections.
Reply: Thanks for your comment. The section title has been revised and highlighted in red.
- Comment: Coefficient of Variation Analysis. This is a basic statistical parameter that indicates the level of variability. It is generally accepted that a CV of up to 10% indicates low variability, 11-20% indicates moderate variability, and above 20% indicates high variability. Numerous studies have also shown that under stress conditions, variability increases. This is primarily due to sample heterogeneity, that is, the different ability of plants within a single genotype to respond to a stress factor. Why did the authors choose to use this parameter for discussion in the text of the article? It is usually not given such close attention. It is more appropriate to analyze mean or median trait values in detail, rather than CV.
Reply: Thanks for your comment. The CV description was changed to the mean value.
- Comment: A clarification about the dots should be added to the caption for Figure 4 – are they outliers or extremes?
Reply: Thanks for your comment. The explains had been added in the Fig. 4 legend.
- Comment: The missing «*», «**», and «***» in Figure 5 should be added, as the captions are present.
Reply: Thanks for your comment. In Fig. 5, paired t-test was performed between the control and waterlogging groups. Only the data with significant differences were marked with asterisks.
- Comment: It is unclear what is meant by GA3? The article lacks a description of one of the GA types (GA3) studied in the work. The list of abbreviations only includes GA.
Reply: Thanks for your comment. Because GA₃ has been the most extensively applied and studied, there was a slight confusion but this mistake has been corrected in the manuscript.
- Comment: In all figures where 8 genotypes are indicated, it is necessary to indicate which of them are sensitive and which are resistant. This will facilitate the understanding of the material.
Reply: Thanks for your comment. We has added some details in Fig 3,4,5,6 and their legends to separate the sensitive and tolerant groups.
- Comment: In the introduction and discussion, the authors discuss the importance of aerenchyma formation under hypoxic conditions, but they do not study this process themselves. How is aerenchyma formation related to salicylic acid synthesis, especially during the initial stages of plant development? Also of interest in studying aerenchyma formation is the analysis of the expression of the β2 proteasome subunit (PBB gene) under hypoxic conditions.
Reply: Thanks for your comment. The formation of aerenchyma in plant roots under waterlogging stress is one of the crucial adaptive traits for plants to cope with hypoxia stress, and numerous studies have reported this phenomenon in crops such as barley, wheat, and rice. Therefore, the research on aerenchyma is mentioned in the background section of this paper. However, the focus of the present study is on the development of adventitious roots induced by waterlogging stress and its regulatory mechanisms, a research field that remains relatively underexplored. The salicylic acid and PBB genes mentioned by the reviewers may be further investigated in subsequent studies.
- Comment: The authors conclude that root growth inhibition in sensitive genotypes depends on ABA and ethylene, i.e., only on the parameters they studied. Data showed that ABA levels decreased in these genotypes. However, the classic role of ABA as a stress hormone would suggest an increase.
Reply: Thanks for your comment. The effect of ABA on root system development exhibits a significant concentration dependence: Low concentrations of ABA (usually referring to the basal level before stress or that induced by mild stress): promote root growth, enhance root water absorption capacity, and improve drought resistance. High concentrations of ABA (accumulated in large quantities under severe stress): inhibit root growth, restrict taproot elongation and lateral root germination, but contribute to resource conservation.
Under waterlogging stress, the ABA concentration in plant roots generally shows a decreasing trend, which is opposite to the typical response of a significant increase in ABA under drought stress. Most plant species exhibit this downward trend; for example, the ABA concentration in soybean roots decreases rapidly by 50% within 24 hours after submergence, and secondary aerenchyma forms after 72 hours (Pan et al; 2021). In wheat roots, ABA concentration displays a dynamic change characterized by a transient increase followed by a continuous decrease under waterlogging conditions (Nan et al; 2002).
Pan J, Sharif R, Xu X and Chen X (2021) Mechanisms of Waterlogging Tolerance in Plants: Research Progress and Prospects. Front. Plant Sci. 11:627331. doi: 10.3389/fpls.2020.627331
Nan, R., Carman, J. G., and Salisbury, F. B. (2002). Water stress, CO2 and photoperiod influence hormone levels in wheat. J. Plant Physiol. 159, 307–312. doi: 10.1078/0176-1617-00703

Reviewer 3 Report
Comments and Suggestions for Authors
Section 1 (Introduction), Lines 39–59: The first introductory paragraph provides a broad overview of waterlogging and its global economic impact, but it reads more like a general review and less like a focused lead-in to this specific barley study. Please shorten this paragraph slightly and end it with a clearer transition that narrows from general flooding damage to the specific problem of waterlogging in barley and its roots under hypoxia.
Section 1 (Introduction), Lines 60–145: The introduction is rich in literature but somewhat long and partially repetitive, especially where similar waterlogging effects and aerenchyma/adventitious root formation in cereals are described multiple times. Consider condensing overlapping descriptions (e.g., root aerenchyma formation and yield losses across cereals) and keeping only the most directly relevant examples to maintain a clear line of argument towards barley and root traits.
Section 1 (Introduction), Lines 168–178: The final paragraph nicely summarizes the study but does not explicitly state the key hypotheses or research questions. To guide the reader, please add one or two explicit sentences such as “We hypothesized that…” or “The main objectives were…” highlighting (i) genotypic differences in waterlogging tolerance across growth stages, and (ii) the expected relationships among root traits, hormone levels, and root development genes.
Section 2.1 (Results), Lines 182–188 and Section 4.1 (Materials and Methods), Page 12–13, Lines 427–431: The term “lose ratio” and its formula ((control−waterlogging)/control) are potentially confusing, particularly because negative values correspond to increased trait values under stress. Please consider renaming this index to a clearer term such as “relative change (%)” or “relative difference (%)”, and ensure the formula and sign convention are clearly explained in both the text and Figure 1 caption.
Section 2.2 (Results), Lines 200–209: The description of coefficients of variation (CV) under control and waterlogging conditions is useful, but the biological interpretation is only briefly mentioned (e.g., “waterlogging severely affected yellow leaf number at three-leaf stage”). It would improve clarity if you briefly explain how higher CVs in SPAD/leaf number relate to genotypic divergence in tolerance and whether specific genotypes drive these differences.
Section 4.2 (Materials and Methods – Waterlogging treatments), Lines 434–451: The pot experiment design is generally clear, but for reproducibility and interpretation, please explicitly state the number of biological replicates used for each measured trait (SPAD, yellow/green leaf number, plant height, spike length, kernels per spike) per genotype and treatment, and confirm whether all six plants per pot or a subsample were used in the measurements.
Section 4.4 (Materials and Methods – Hormone contents measurements), Lines 469–475: The hormone assay description currently cites only the commercial kits. Please add key methodological details, including (i) fresh or dry weight of root tissue used per sample, (ii) brief extraction protocol (solvent, incubation, centrifugation), (iii) units in which ABA, GA and ACC are expressed (e.g., ng g⁻¹ FW), and (iv) whether standard curves or internal standards were employed. These details are important for reproducibility and comparison with other studies.
Section 4.6 (Materials and Methods – Statistical analysis), Lines 490–494: You state that one-way ANOVA followed by Tukey’s test was used, but it is not indicated whether the assumptions of ANOVA (normality and homogeneity of variance) were checked. Please clarify how assumptions were verified and, if necessary, whether transformations were applied to the data.
Section 2.5 and 2.6 (Results – Hormones and Gene Expression), Lines 272–284 and Lines 290–303; Section 3 (Discussion), Lines 394–407; Section 5 (Conclusions), Lines 505–515: The data on hormones (ABA, GA3, ACC) and expression of TCP20, PLT2, SHY2, SHR, SCR, PILS2 are based on a single time point and show genotype-dependent patterns. Some sentences in the Discussion and Conclusions interpret these correlations as causal mechanisms (e.g., GA/ethylene “thereby inducing” SHR and SCR to promote root growth; ABA “inducing” SHY2 to inhibit root growth in sensitive genotypes). I recommend softening the language (e.g., “our results are consistent with the hypothesis that…”) and clearly acknowledging that functional validation (mutants, overexpression, or time-course analyses) would be required to prove these regulatory relationships in barley.
Section 3 (Discussion), Lines 394–397: The statement “we suggested that the waterlogging-tolerant barley genotypes respectively induce and inhibit the positive and negative regulated gene expressions that control the meristem at the root tips” is quite general and implies a unified regulatory pattern. Since Figure 6 shows genotype-specific and sometimes inconsistent expression changes, please rephrase this to emphasize that these patterns are suggestive trends rather than a universally shared mechanism among all tolerant vs. sensitive genotypes.
Figures and Tables, Figure 1; Figure 3, Figure 4; Figure 5; and Table 1, Table 2: The figures and tables are informative but can be made clearer. Please (i) increase the font size of axis labels and legends to improve readability, (ii) ensure all trait units (e.g., mm, mm², %, ng g⁻¹) are explicitly shown, and (iii) define all abbreviations (MFV, HWT, WT, MWS, WS, TLS, JS, BS, SPAD) in each table caption and at first use in the text.
Comments on the Quality of English LanguageQuality of English Language (multiple sections), e.g. Lines 25–27; Line 61; Line 155; Lines 371–377; Lines 505–513: The manuscript is generally understandable but contains numerous language issues, including non-standard word choices (“etcetera traits”), awkward or incorrect phrases (“From now, 48 quantitative trait loci…”, “chagned”), and several very long sentences that could be split for clarity. I strongly recommend a thorough language revision by a fluent English speaker or professional editing service to correct grammar, spelling, and syntax, replace informal terms (e.g., repeated “etcetera”) with precise scientific wording, and improve overall readability.
Author Response
Author Response to Reviewer Comments
- Comment: Section 1 (Introduction), Lines 39–59: The first introductory paragraph provides a broad overview of waterlogging and its global economic impact, but it reads more like a general review and less like a focused lead-in to this specific barley study. Please shorten this paragraph slightly and end it with a clearer transition that narrows from general flooding damage to the specific problem of waterlogging in barley and its roots under hypoxia.
Reply: Thanks for your comment. The first paragraph of introduction has been shortened and revised.
- Comment: Section 1 (Introduction), Lines 60–145: The introduction is rich in literature but somewhat long and partially repetitive, especially where similar waterlogging effects and aerenchyma/adventitious root formation in cereals are described multiple times. Consider condensing overlapping descriptions (e.g., root aerenchyma formation and yield losses across cereals) and keeping only the most directly relevant examples to maintain a clear line of argument towards barley and root traits.
Reply: Thanks for your comment. According to your suggestions, the review content irrelevant to barley waterlogging stress in the introduction section has been deleted and revised.
- Comment: Section 1 (Introduction), Lines 168–178: The final paragraph nicely summarizes the study but does not explicitly state the key hypotheses or research questions. To guide the reader, please add one or two explicit sentences such as “We hypothesized that…” or “The main objectives were…” highlighting (i) genotypic differences in waterlogging tolerance across growth stages, and (ii) the expected relationships among root traits, hormone levels, and root development genes.
Reply: Thanks for your comment. The explicit sentences have been added “In this study, the main objectives were (i) genotypic differences in waterlogging toler-ance across whole growth stages to screen representative waterlogging-tolerant and waterlogging-sensitive barley genotypes; (ii) effects of the early stage waterlogging stress on root morphology, hormone content and relative gene expression to explore the mechanism of waterlogging stress tolerance in barley”.
- Comment: Section 2.1 (Results), Lines 182–188 and Section 4.1 (Materials and Methods), Page 12–13, Lines 427–431: The term “lose ratio” and its formula ((control−waterlogging)/control) are potentially confusing, particularly because negative values correspond to increased trait values under stress. Please consider renaming this index to a clearer term such as “relative change (%)” or “relative difference (%)”, and ensure the formula and sign convention are clearly explained in both the text and Figure 1 caption.
Reply: Thanks for your comment. The term “lose ratio” has been changed to “relative changes” and explained in the text and caption.
- Comment: Section 2.2 (Results), Lines 200–209: The description of coefficients of variation (CV) under control and waterlogging conditions is useful, but the biological interpretation is only briefly mentioned (e.g., “waterlogging severely affected yellow leaf number at three-leaf stage”). It would improve clarity if you briefly explain how higher CVs in SPAD/leaf number relate to genotypic divergence in tolerance and whether specific genotypes drive these differences.
Reply: Thanks for your comment. We got one reviewer’s comment that “CV parameter is usually not given such close attention and more appropriate to analyse mean trait values”. So, in this result part we changed to describ mean value.
- Comment: Section 4.2 (Materials and Methods – Waterlogging treatments), Lines 434–451: The pot experiment design is generally clear, but for reproducibility and interpretation, please explicitly state the number of biological replicatesused for each measured trait (SPAD, yellow/green leaf number, plant height, spike length, kernels per spike) per genotype and treatment, and confirm whether all six plants per pot or a subsample were used in the measurements.
Reply: Thanks for your comment. The explains have been added “During the measurement, totally 12 biological replicates were used for all the parame-ters including SPAD, plant height, the number of yellow or green leaves, the kernels per spike and spike length measurements per genotype and treatment”.
- Comment: Section 4.4 (Materials and Methods – Hormone contents measurements), Lines 469–475: The hormone assay description currently cites only the commercial kits. Please add key methodological details, including (i) fresh or dry weight of root tissue used per sample, (ii) brief extraction protocol (solvent, incubation, centrifugation), (iii) units in which ABA, GA and ACC are expressed (e.g., ng g⁻¹ FW), and (iv) whether standard curves or internal standards were employed. These details are important for reproducibility and comparison with other studies.
Reply: Thanks for your comment. The details have been added “Briefly, after the stress treatment, root tissues were collected, weighed, and immedi-ately preserved in liquid nitrogen. For detection, tissue extraction solution was added to the preserved samples, followed by grinding. The homogenate was centrifuged at 2000 rpm for 20 minutes, and the supernatant was collected for subsequent assays us-ing commercial kits. A standard curve was generated with the standard substances provided in the kit, and the hormone content of the experimental samples was calcu-lated based on their detected values, with the unit expressed as pmol/L”.
- Comment: Section 4.6 (Materials and Methods – Statistical analysis), Lines 490–494: You state that one-way ANOVA followed by Tukey’s test was used, but it is not indicated whether the assumptions of ANOVA (normality and homogeneity of variance) were checked. Please clarify how assumptions were verified and, if necessary, whether transformations were applied to the data.
Reply: Thanks for your comment. The Shapiro-Wilk test was performed on the raw data with SPSS software, and the results showed that P > 0.05, indicating that the data conformed to a normal distribution. Subsequently, the test of Homogeneity of variances was also conducted with SPSS software. The results demonstrated that the significance values were greater than 0.05 which indicated the data were suitable for one-way ANOVA analysis.
- Comment: Section 2.5 and 2.6 (Results – Hormones and Gene Expression), Lines 272–284 and Lines 290–303; Section 3 (Discussion), Lines 394–407; Section 5 (Conclusions), Lines 505–515: The data on hormones (ABA, GA3, ACC) and expression of TCP20, PLT2, SHY2, SHR, SCR, PILS2 are based on a single time point and show genotype-dependent patterns. Some sentences in the Discussion and Conclusions interpret these correlations as causal mechanisms (e.g., GA/ethylene “thereby inducing” SHR and SCR to promote root growth; ABA “inducing” SHY2 to inhibit root growth in sensitive genotypes). I recommend softening the language (e.g., “our results are consistent with the hypothesis that…”) and clearly acknowledging that functional validation (mutants, overexpression, or time-course analyses) would be required to prove these regulatory relationships in barley.
Reply: Thanks for your comment. We have polished the language and enhanced the inferential tone in accordance with your comments. All these revised parts have been marked in red in the manuscript.
- Comment: Section 3 (Discussion), Lines 394–397: The statement “we suggested that the waterlogging-tolerant barley genotypes respectively induce and inhibit the positive and negative regulated gene expressions that control the meristem at the root tips” is quite general and implies a unified regulatory pattern. Since Figure 6 shows genotype-specific and sometimes inconsistent expression changes, please rephrase this to emphasize that these patterns are suggestive trends rather than a universally shared mechanism among all tolerant vs. sensitive genotypes.
Reply: Thanks for your comment. This sentence has been changed to “Based on these results, we proposed a hypothesis that most of waterlogging-tolerant barley genotypes may respectively induce and inhibit the positive and negative regu-lated gene expressions that control the meristem at the root tips to cope with water-logging stress. The further functional validation experiment is needed and explored in future research”.
- Comment: Figures and Tables, Figure 1; Figure 3, Figure 4; Figure 5; and Table 1, Table 2: The figures and tables are informative but can be made clearer. Please (i) increase the font size of axis labels and legends to improve readability, (ii) ensure all trait units (e.g., mm, mm², %, ng g⁻¹) are explicitly shown, and (iii) define all abbreviations (MFV, HWT, WT, MWS, WS, TLS, JS, BS, SPAD) in each table caption and at first use in the text.
Reply: Thanks for your comment. All the suggestions were adopted and modifications were made in the manuscript and figures.
- Comment: Comments on the Quality of English Language
Quality of English Language (multiple sections), e.g. Lines 25–27; Line 61; Line 155; Lines 371–377; Lines 505–513: The manuscript is generally understandable but contains numerous language issues, including non-standard word choices (“etcetera traits”), awkward or incorrect phrases (“From now, 48 quantitative trait loci…”, “chagned”), and several very long sentences that could be split for clarity. I strongly recommend a thorough language revision by a fluent English speaker or professional editing service to correct grammar, spelling, and syntax, replace informal terms (e.g., repeated “etcetera”) with precise scientific wording, and improve overall readability.
Reply: Thanks for your comment. We have invited experts to conduct a rigorous linguistic revision of this manuscript.

Round 2
Reviewer 2 Report
Comments and Suggestions for Authors
The authors responded to all of the expert's comments and slightly revised the article. However, it is unclear why the authors repeatedly highlighted in red text that remained unchanged from the first version of the article.
In addition, I have a number of new recommendations:
1) Line 33. Remove the hyphen in the word "gen-otypes";
2) Line 50. Latin names are usually written in parentheses, so I propose replacing "Hordeum vulgare L. (barley)" with "Barley (Hordeum vulgare L.)";
3) Line 156. Remove the hyphen in the word "con-trol";
4) Add a period after the abbreviation "Fig" throughout the article;
5) Table 1. All abbreviations must be expanded;
6) Table 2. Replace "WHT" with "HWT";
7) Line 517. Replace "waterlogging-sensitive" with "moderate waterlogging-sensitive".
Author Response
Author Response to Reviewer Comments
1) Comment :Line 33. Remove the hyphen in the word "gen-otypes";
Reply: Thanks for your comment. The mistake had been corrected.
2) Comment : Line 50. Latin names are usually written in parentheses, so I propose replacing "Hordeum vulgare L. (barley)" with "Barley (Hordeum vulgare L.)";
Reply: Thanks for your comment. This sentence had been changed to “Barley (Hordeum vulgare L.)….”
3) Comment : Line 156. Remove the hyphen in the word "con-trol";
Reply: Thanks for your comment. The mistake had been corrected.
4) Comment :Add a period after the abbreviation "Fig" throughout the article;
Reply: Thanks for your comment. The dot had been added after the Fig.
5) Comment :Table 1. All abbreviations must be expanded;
Reply: Thanks for your comment. The full names were added in the table note.
Note: SPAD: soil and plant analyzer development, relative chlorophyl content; Con: control; WL: waterlogging; no.: number; CV: variation coefficients; T x C: treatment vs control.
6) Comment :Table 2. Replace "WHT" with "HWT";
Reply: Thanks for your comment. The mistake had been corrected.
7) Comment :Line 517. Replace "waterlogging-sensitive" with "moderate waterlogging-sensitive".
Reply: Thanks for your comment. The mistake had been corrected.
